# Is Accurate Synoptic Altimetry Achievable by Means of Interferometric GNSS-R?

**Fran Fabra** [1,2,*], **Estel Cardellach** [1,2], **Serni Ribó** [1,2], **Weiqiang Li** [1,2], **Antonio Rius** [1,2], **Juan Carlos Arco-Fernández** [1,2,†], **Oleguer Nogués-Correig** [1,2,‡], **Jaan Praks** [3], **Erkka Rouhe** [3], **Jaakko Seppänen** [3,§] **and Manuel Martín-Neira** [4]

[1]  Earth Observation Research Group, Institute of Space Sciences (ICE, CSIC), 08193 Barcelona, Spain; estel@ice.csic.es (E.C.); ribo@ice.csic.es (S.R.); weiqiang@ice.csic.es (W.L.); rius@ice.csic.es (A.R.); juancarlos_arco@hotmail.com (J.C.A.-F.); oleguer@gmail.com (O.N.-C.)

[2]  Institut d'Estudis Espacials de Catalunya (IEEC), 08034 Barcelona, Spain

[3]  Department of Electronics and Nanoengineering, Aalto University, 02150 Espoo, Finland; jaan.praks@aalto.fi (J.P.); erkka.rouhe@aalto.fi (E.R.); jaakko.seppanen@fmi.fi (J.S.)

[4]  European Space Research and Technology Centre, European Space Agency, 2200 AG Noordwijk, The Netherlands; Manuel.Martin-Neira@esa.int

*   Correspondence: fabra@ice.csic.es; Tel.: +34-93-737-9788 (ext. 933031)

†   Current address: ERNI Consulting Spain, 08002 Barcelona, Spain.

‡   Current address: Spire Global UK Limited, Glasgow G3 8JU, UK.

§   Current address: Finnish Meteorological Institute, FI-00101 Helsinki, Finland.

**Abstract:** This paper evaluates the capability of interferometric global navigation satellite system reflectometry (GNSS-R) to perform sea surface altimetry in a synoptic scenario. Such purpose, which requires the combination of the results from different GNSS signals, constitutes a unique characteristic of this approach. Interferometric GNSS-R group delay altimetry has been proven to be more precise than conventional GNSS-R. However, the self-consistency and accuracy of their synoptic solutions (simultaneous multi-static results) have never been proved before. In our work, we analyze a dataset of GNSS signals reflected off the Baltic Sea acquired during an airborne campaign using a receiver that was developed for such a purpose. Among other features, it enables beamformer capability in post-processing to get multiple and simultaneous GNSS signals under the interferometric approach's restrictions. In particular, the signals from two GPS and two Galileo satellites, at two frequency bands (L1 and L5), covering an elevation range between 28° and 83°, are processed to retrieve sea surface height estimations. The results obtained are self-consistent among the different GNSS signals and data tracks, with discrepancies between 0.01 and 0.26 m. Overall, they agree with ancillary information at 0.40 m level, following a characteristic height gradient present at the experimental site.

**Keywords:** GNSS-R; altimetry; interferometry; radar; GPS; Galileo; sea level

## 1. Introduction

The use of signals of opportunity represents a low cost means towards remote sensing of geophysical parameters. In particular, the approach known as global navigation satellite system reflectometry (GNSS-R) or passive reflectometry and interferometry system (PARIS) [1] has been tested for a wide variety of applications during the previous years (a comprehensive list of examples can be found in [2,3] or [4]). Being benefited by the increasing GNSS constellation, this technique provides gaped wide swath coverage and thus enables synoptic views under all weather conditions, as L-band signals penetrate thick clouds and precipitation cells. All these aspects have motivated the deployment of dedicated GNSS-R receivers onboard experimental satellite platforms, such as the Disaster Monitoring Constellation satellite [5] and TechDemoSat-1 (TDS-1) [6], and an operational mission like

NASA's cyclone global navigation satellite system (CYGNSS) [7], composed of eight micro-satellites. Opportunistic GNSS reflected signals were also obtained aboard the spaceborne imaging radar-C (SIR-C) [8] and the receiving chain of the soil moisture active passive (SMAP) mission [9].

Although the aforementioned spaceborne systems were designed for scatterometry applications as a primary target, several studies show the potential of the GNSS-R technique for altimetry from their datasets (e.g., [10–16]). In fact, as it was initially envisaged in [1], the multi-static geometry nature of this approach represents a good opportunity towards ocean altimetry. The reason behind this is that the enhanced spatio-temporal coverage achieved would enable the monitoring and forecasting of mesoscale ocean signals (range of 30–300 km), thus complementing other altimetric products provided by monostatic Radar. Moreover, compared against other signals of opportunity, GNSS-R takes advantage of a constellation of transmitters that has to be properly maintained, on a long term basis, and with well defined orbits in order to guarantee its main target of offering a reliable positioning service. The last point is especially relevant for altimetry, which is based on delay observables. On the other hand, the characteristics of the GNSS signals, which are not optimized for these types of applications, impose certain constraints to the precision that can be achieved. In particular, the transmitter power is relatively weak, since it is designed to provide raw signal levels below the noise floor when reaching line-of-sight targets (with a standard antenna) located at the Earth surface, and therefore does not take into account the signal-to-noise ratio (SNR) degradation after reflection. In addition, the open-access modulation codes have much narrower bandwidths than those transmitted by conventional radar altimeters or other sources of opportunity, such as digital TV signals broadcast by geostationary satellites (e.g., [17,18]). In order to overcome these limitations, an alternative approach consists in cross-correlating both raw direct and reflected signals. Unlike conventional GNSS-R, this method takes profit of all codes present in the GNSS signal regardless of their encryption status, thus increasing the effective bandwidth employed. For this reason, this approach, known as interferometric GNSS-R (iGNSS-R), was selected for different European Space Agency's proposed GNSS-R altimetric missions: the PARIS in-orbit demonstrator (PARIS IoD) [19], the GNSS REflectometry, radio occultation, and scatterometry onboard the International Space Station (GEROS-ISS) [20] and the GNSS transpolar Earth reflectometry exploring system (G-TERN) [21].

Within this context, we developed the first iGNSS-R dedicated instrument, the PARIS interferometric receiver (PIR), and tested its performance from an 18 m-high bridge over estuary waters [22]. Then, the next step consisted in adapting the receiver for an airborne platform (PIRA) to carry out an experimental campaign for altimetry over open waters. The analysis of the dataset, acquired onboard an aircraft flying at a mean altitude of 3 km, enabled a more realistic assessment of the altimetric precision of the iGNSS-R concept [23]. The analysis was limited to a single GPS reflection collected during two tracks of about 15 min each at a high elevation (>80°). The reason behind such limitation was related to one of the main constraints of the iGNSS-R concept: it is not possible to separate different GNSS signals by means of their modulation codes. Since neither range nor Doppler-separability were a valid option in that particular scenario (receiver's velocity and height were relatively low compared to a low Earth orbit platform), the strategy followed was to use antenna arrays with high directivity and then to select the proper time interval to acquire a single GNSS contribution. Therefore, to prove the synoptic capability of the iGNSS-R concept by combining the results from several GNSS signals at different geometries was out of the scope at that time. By synoptic altimetry it is meant performing ocean altimetry observations along several tracks simultaneously covering a broad area. Such combinations of different reflected signals is the key aspect from GNSS-R to become an useful tool to complement other altimetric products provided by conventional Radar, yet the consistency and accuracy of the multiple simultaneous (synoptic) altimetric observations had never been proved. In other words, is the performance degraded when implemented into a multiple-beam dynamic steering system? Are the multiple solutions consistent (reliable) or do they have intrinsic biases or other non-controlled effects? These blocks of questions have never been tackled with actual data and they represent key pieces to assess the suitability of the concept in possible future spaceborne scenarios.

This paper describes the work done towards the fulfilment of that last requirement: to check the capability of iGNSS-R to perform multi-static sea surface altimetry by keeping the same accuracy levels previously obtained. Section 2 gives details about the new receiver deployed and the airborne experimental campaign carried out to acquire the proper dataset. Then, Section 3 describes the methodology followed to get the altimetric results, which are shown and discussed in Section 4. Finally, Section 5 provides some conclusions about the work done.

## 2. SPIR Campaign

Once a proper GNSS-R altimetric precision has been proved based on the interferometric technique applied towards single and isolated transmitters (PIRA campaign 2011 [23]), the next step is to prove the synoptic capability of this technique given the available and increasing GNSS constellation. For this reason, a new flight campaign was later carried out on 3rd December 2015 under similar conditions to 2011 to monitor the ellipsoidal height gradient over the Baltic Sea near Helsinki (Finland), as shown in the flight path provided in Figure 1. The key difference was the use of a new instrument developed by Institut d'Estudis Espacials de Catalunya (IEEC) for the acquisition of GNSS-R signals: the software PARIS interferometric receiver (SPIR) [24]. Unlike its real-time instrument predecessors (GOLD-RTR and PIR/A), SPIR is a high-speed recording receiver that collects raw complex GNSS signals at 80 MHz of sampling rate (1-bit—in-phase and quadrature—quantization) from 16 front-ends that can operate at any of the common GNSS L1, L2 and L5 bands. Two eight-element antennas were installed at Aalto's Skyvan aircraft (roof and belly locations) to feed the 16 SPIR's RF inputs for the acquisition of both up- and down-looking signals, as shown in Figure 2. Such architecture enables, among other possibilities, beamforming capability in post-processing, which is a relevant aspect to properly investigate the processing steps and final performance of iGNSS-R synoptic altimetry. The main reason is that a proper separability between different GNSS signals is required for obtaining clean waveforms under the interferometric approach, since unlike in a satellite-base scenario (such as in [19–21]), the relatively low height and velocity of the aircraft platform does not offer a significant separation in range or Doppler among the available GNSS transmitters. Given that code, time and frequency-based signal multiplexing are also discarded, we have to mostly rely on antenna directivity to achieve such separability. These concepts will be later illustrated with real examples.

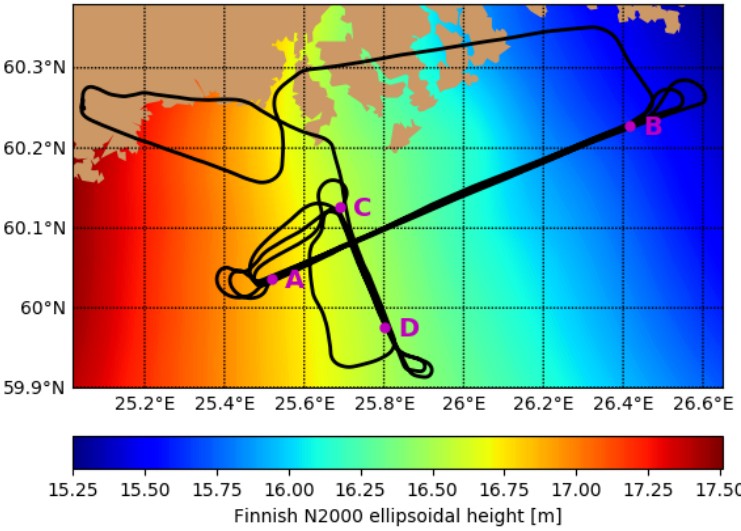

**Figure 1.** Flight path of the experimental campaign over the Baltic Sea. The background color of the sea indicates the ellipsoidal heights from Finnish N2000 system according to FIN2005N00 conversion surface [25]. The location of four reference waypoints (A, B, C and D) is indicated.

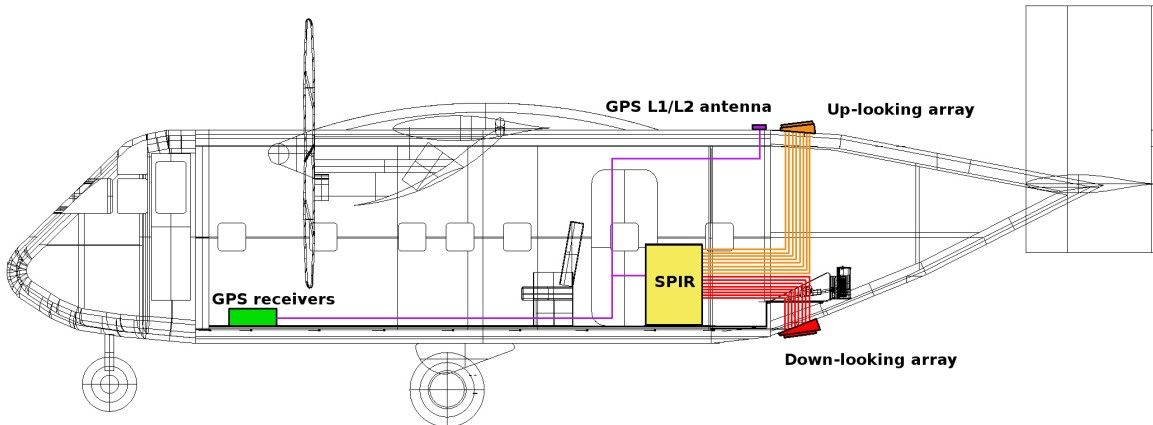

**Figure 2.** Location of main instrumental setup onboard Skyvan aircraft. Software passive reflectometry and interferometry system (PARIS) interferometric receiver (SPIR) is fed by 16 individual antenna elements, eight from the antenna array pointing to the zenith (orange) and eight from the antenna array pointing to the nadir (red). Ancillary GPS receivers and avionic antenna for navigation and time tagging (green and purple respectively) were also used, together with an inertial system synchronized to the GPS receivers (not depicted).

At this point, the reader will note that we were seeking two contradictory conditions during the planning of the campaign: (1) to have several GNSS reflections at the same time but (2) properly separated to achieve the best isolation among them for a wide elevation range. The achievable directivity is limited by the number of antenna elements in the arrays. Therefore, the time window of the campaign was carefully selected based on estimations of GNSS visibility after taking into account the trade-off between maximum number of simultaneous observations and their angular separability. Figure 3 provides the actual GNSS visibility seen from the aircraft during the flight. We can see that the scenario was dominated by three GPS (PRN 01, 03 and 32) and two Galileo (PRN 11 and 19) results. However, GPS with PRN32 corresponded to a block II-A satellite at that time, without the broadest M-code or L5 signal, so it was finally discarded from the altimetric analysis.

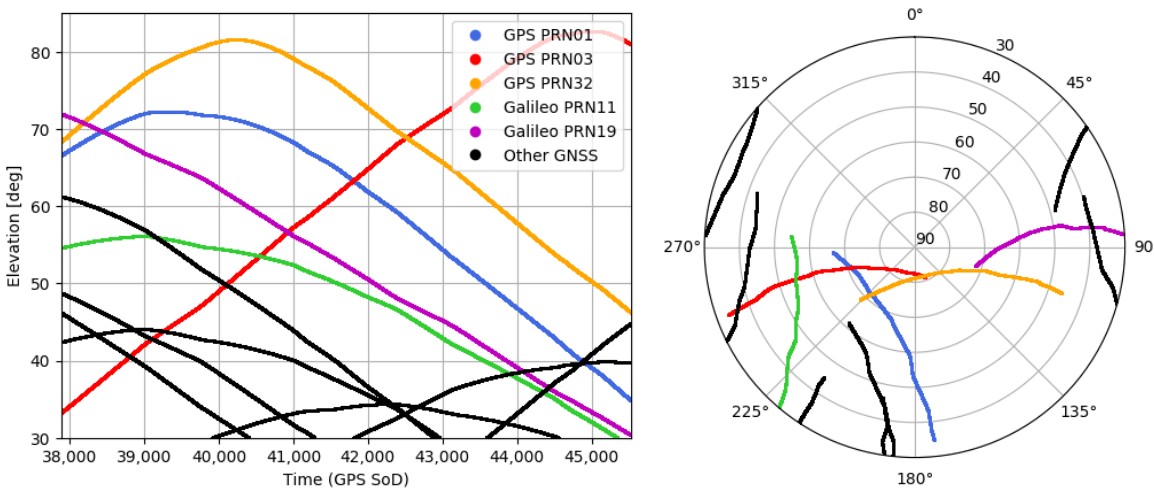

**Figure 3.** Global navigation satellite system (GNSS) visibility as seen from the receiver during the flight. (**Left**) Evolution of the elevation angle with time. (**Right**) Polar view as a function of azimuth and elevation angles.

As previously displayed in Figure 1, the flight consisted in a set of passes between two pairs of waypoints (AB and CD). Their location, as it happened in the 2011 PIRA campaign, was selected to have two straight flight trajectory intervals: parallel (AB) and perpendicular (CD) to the ellipsoidal

height gradient of the sea surface. Table 1 provides the most relevant information about the different flight segments. As we can see, during the first segment (B-to-A), the SPIR was configured to collect data at L5 band (as a secondary objective). After that, the receiver was changed to L1-acquisition (primary objective). The remaining flight path consisted in two perpendicular trajectories, which were travelled in both senses (A-to-B, B-to-A, C-to-D and D-to-C). This scheme was done two times, resulting then in eight segments at L1. During all of them, the nominal height of the receiver was around 3 km. In order to avoid uncertainties caused by aircraft maneuvers or significant altitude variations hindering the beamforming, the data analysis and results shown in this paper are limited to these straight flight segments. In order to differentiate these flight intervals with their corresponding datasets acquired for each particular GNSS satellite and PRN, we will use the term track when referring to later ones in the follow-on analysis. In particular, during every flight segment, four tracks (from GPS PRN01, GPS PRN03, Galileo PRN11 and Galileo PRN19) will be analyzed.

**Table 1.** Relevant information on the different flight segments. Two different time references are given: GPS second of the day (SoD) and time from connected segments (TCS) after removing aircraft maneuvers.

| Segment Label | Start Time [GPS SoD] | Start Time [TCS sec] | Distance [km] | Mean Speed [m/s] | Freq. Band |
|---|---|---|---|---|---|
| 0-BA | 37,890 | 0 | 56.2 | 57.3 | L5 |
| 1-AB | 39,120 | 990 | 49.9 | 96.0 | |
| 2-BA | 39,890 | 1520 | 56.7 | 54.5 | |
| 3-CD | 41,220 | 2570 | 16.1 | 80.7 | |
| 4-DC | 41,630 | 2780 | 18.3 | 63.1 | L1 |
| 5-AB | 42,480 | 3080 | 50.5 | 100.9 | |
| 6-BA | 43,280 | 3590 | 58.4 | 50.8 | |
| 7-CD | 44,830 | 4750 | 17.0 | 80.8 | |
| 8-DC | 45,200 | 4970 | 19.9 | 64.3 | |

Finally, different ancillary datasets were collected during the campaign, being the most relevant for this study the aboard dual-band GNSS information for the precise positioning of the aircraft trajectory, the inertial information provided by the inertial measurements unit (IMU) to determine the relative location and orientation of the antenna arrays, and wind speed and sea level estimations taken from buoys and nearby meteorological stations provided by the Finnish Meteorological Institute. The precise trajectory was computed by [26], whose standard deviation range between 3 and 5 cm during the segments analyzed. From the last measurements we obtain, for a given time (here-on referred as $t$) and location $[\lambda, \phi]$ (longitude and latitude) of a GNSS signal's specular point of reflection, our reference sea surface height (SSH) estimation as:

$$\text{SSH}_{\text{ref}}[t, \lambda, \phi] = h_{\text{FIN}}[\lambda, \phi] + \Delta h_{\text{mean}}[\lambda, \phi] + h_{\text{sea}}[t, \lambda, \phi], \tag{1}$$

where $h_{\text{FIN}}$ refers to the FIN2005N00 height conversion surface for Finland [25]. This model provides the normal height offset between Finnish N2000 height system and the European terrestrial reference system 1989 (ETRF89). The last one is based on the geodetic reference system 1980 (GRS80), which compared against our reference ellipsoid, world geodetic system 84 (WGS84), just differs 0.105 mm in the semi polar axis. The next term, $\Delta h_{\text{mean}}$, is the normal height offset between Finnish N2000 height system and the theoretical mean sea level estimated by the Finnish Meteorological Institute [27]. For 2015, this value was 0.200 m at Helsinki and 0.201 m at Porvoo, located approximately at both ends of the flight path. The last term, $h_{\text{sea}}$, is the sea surface height measured by those buoys with respect to mean sea level. Two time series were obtained from the two previous locations (Helsinki and Porvoo) at a rate of 30 minutes. The procedure to determine the contribution of $\Delta h_{\text{mean}} + h_{\text{sea}}$ consists in computing the values of $h_{\text{sea}}$ for both locations at $t$ by means of a spline interpolation, and then to

compute the weighted mean of both values based on the linear distance of $[\lambda, \phi]$ with respect to them. The difference between the solutions from both locations ranges from 1 to 3 cm during the experiment, so the selected procedure should not have an important impact on the results.

Then, as a summary after taking into account all these aspects, the objective of the SPIR experimental campaign was to perform sea surface altimetry by means of iGNSS-R from a dynamic platform (aircraft at 3 km height) after combining (1) two frequencies (L1 and L5) and their corresponding signals (with different autocorrelation function); (2) two GNSS systems (GPS and Galileo); and (3) several transmitters at the same time with different geometries (two PRN for each system). This combination represents a unique study in the field.

## 3. Data Processing and Inversion

### 3.1. From Raw Signals to Power Waveforms

Once the raw samples from both up- and down-looking antenna arrays were collected, summing up 3 TB of data, the next processing steps were done offline. Figure 4 provides a schematic flow chart of the whole process. First of all, the contributions from each element of the arrays are combined in order to coherently maximize the signal amplitude for a given direction, or in other words, to steer the main beam of the array factor towards that direction (beamforming process). With the target of GNSS-R altimetry, we point the up-looking array towards a desired GNSS transmitter and the down-looking one towards its corresponding specular point estimated over ellipsoid WGS84, as described in [24]. These computations required the precise position of the aircraft, obtained from standard GNSS data processing; the orientation of the arrays, estimated from the inertial measurements unit aboard; and the position of the visible GNSS constellation of transmitters, given by the International GNSS Service.

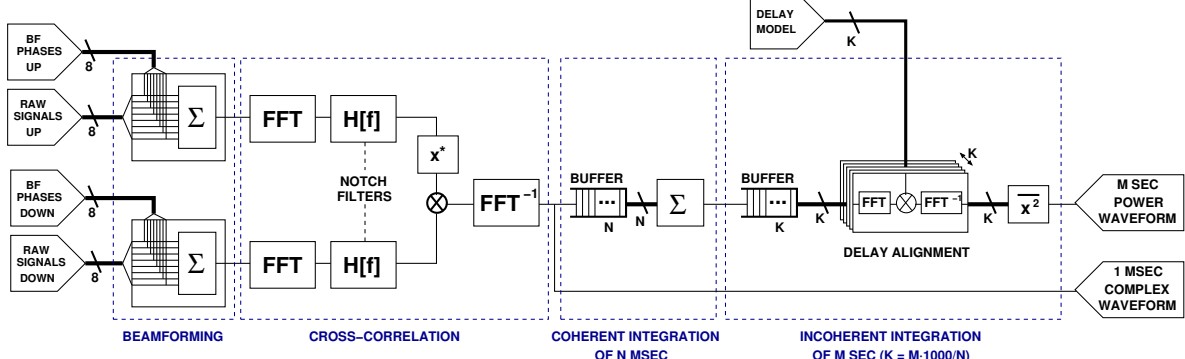

**Figure 4.** Schematic flow chart of the process followed to construct complex and power waveforms from raw signals.

Based on the iGNSS-R approach, the direct and reflected signals previously obtained were then cross-correlated to construct a complex waveform from 1 ms of raw data. This process was implemented in the frequency domain based on Fourier transform principles. Moreover, a set of notch filters were applied with the purpose of mitigating the presence of radio-frequency interferences (RFI), both from the receiver itself or from external sources, whose frequencies were determined after an a-priori dedicated analysis of the spectra from the collected signals. In particular, a strong RFI was located at L1, resulting in an equivalent removal of the C/A-code component for GPS.

Finally, coherent (complex sum) and incoherent (averaging of the squared amplitudes) integrations were performed in order to decrease the impact of thermal and speckle noise respectively in the resultant power waveforms. Before the incoherent integration process, the waveform series were aligned with respect to the first element based on a precise estimation of the delay evolution during the whole time interval. This process tries to minimize the blurring effect that would result in the power waveform shape after averaging misaligned elements, thus causing a degradation in the retrieved

altimetry. In order to avoid sampling rate limitations, this step was done at the frequency domain by means of a product with a phasor rotated with the corresponding delay difference between waveforms.

### 3.2. Delay Model

The precise delay model employed during the alignmnent of the waveform series for a given GNSS signal (here-on referred as $s$) is computed from three components:

$$\rho_{\mathrm{mod}}[t,s] = \rho_{\mathrm{geo}}^{\mathrm{WGS84}}[t,s] - \rho_{\mathrm{ecc}}[t,s] + \rho_{\mathrm{trop}}[t,s]. \tag{2}$$

For the first term, defined as geometric delay $\rho_{\mathrm{geo}}^{\mathrm{WGS84}}$, we characterized our system with the Earth-centered Earth-fixed (ECEF) Cartesian coordinates of the transmitter and receiver and we take ellipsoid WGS84 as a reference model of the Earth's surface. Then, we identified the specular point over such a surface where the reflected signal fulfils Snell's law (the incidence equals the scattering angle). Finally, we computed $\rho_{\mathrm{geo}}^{\mathrm{WGS84}}$ as the difference between the modelled reflected signal's path (transmitter-to-specular-to-receiver) and the direct signal's path (transmitter-to-receiver).

The eccentricity delay $\rho_{\mathrm{ecc}}$ is the correction that has to be applied to $\rho_{\mathrm{geo}}^{\mathrm{WGS84}}$ due to the fact that the up and down-looking antenna arrays were not at the same location. The quantity to compute is the projection of the antenna baseline vector, defined as the relative position between both arrays in the aircraft body frame, into the arrival direction of the reflected signal (we select the up-looking antenna array as the reference). For this purpose, it is required to use the inertial information provided by the IMU to properly rotate the aircraft body frame into the characterized geometrical scenario. The error of this instrument was about 0.6° in tilt, while it ranges from 6° to 8° in heading. When adding this information into the estimation of $\rho_{\mathrm{ecc}}$, it results in an average error of 0.29 cm, with a maximum threshold of 1.75 cm.

Finally, the tropospheric delay $\rho_{\mathrm{trop}}$ takes into account the differential delay increment between the direct and the reflected path (contained in $\rho_{\mathrm{geo}}^{\mathrm{WGS84}}$) when the GNSS signal is travelling across the troposphere. We employed a simple model for its estimation [4]:

$$\rho_{\mathrm{trop}}[t,s] = 2 \frac{2.3}{\sin(\varepsilon[t,s])} \left(1 - e^{-H_{\mathrm{R}}[t]/H_{\mathrm{trop}}}\right), \tag{3}$$

where $\varepsilon$ is the elevation angle, $H_{\mathrm{R}}$ is the receiver's height and $H_{\mathrm{trop}}$ is an estimation of the height of the troposphere at the experiment's location for this time of the year (we take $H_{\mathrm{trop}} = 8621$ km). As commonly done for a scenario with a relatively low-height receiver, the differential effects of the upper layers, such as the ionosphere, were negligible and thus not taken into account.

### 3.3. Waveform Model

Similarly to the procedure proposed in [20], our altimetric inversion was based on a differential approach comparing data against its corresponding model. The main purpose of this approach is to correct deviations in the altimetric inversion caused by effects that may distort the waveform's shape, especially taking into account that our final target will be to check the consistency between different GNSS signals and flight segments. Such effects include: (1) the impact of different codes for each GNSS system and frequency band; (2) the bias contribution of the state of the sea (as analyzed in [28]); (3) the impact of the Doppler bandwidth in the coherent integration for segments with different receiver velocity (both speed and direction); (4) the different weight of the equivalent antenna pattern for changes in elevation and aircraft orientation towards the transmitters; (5) the contribution from a coherent component in the reflected signal; and (6) the impact of the receiver's bandwidth (set to 12 MHz baseband).

Therefore, for every power waveform of data, we build its corresponding model. Our approach was based on [29] and includes the coherent signal term from [30] combined with the roughness effect defined in Equation (3) from [31]. All the aforementioned effects were taken into account during

the simulation of a comprehensive scenario fed by all the ancillary information collected during the campaign. In particular, the sea state was computed from the mean square slope values derived from wind speed and wind direction by means of [32]. For a more realistic computation of the specular point, we added a vertical height to ellipsoid WGS84 corresponding to the value of Finnish N2000 height system (based on FIN2005N00 [25]) interpolated at the a-priori location of the specular point (two-iteration method).

As a last remark, when referring to a waveform model we mean a single simulated GNSS signal, without taking into account additional GNSS contributions.

### 3.4. Primary Observable: Specular Delay

The methodology followed for the retrieval of the specular delay is described in [28], which essentially estimates this value as the maximum of the derivative in the leading edge of the power waveform. It is important to mention that in the analysis shown in [33] with part of the same dataset, we obtained an improvement in the altimetric error by using a model fitting procedure that employs an interval of samples around the specular delay. However, this study was limited to GPS L5 (tracks from GPS PRN01 and PRN03 during flight segment 0-BA), which has a simple autocorrelation function that benefits this methodology. As previously mentioned, the aim of the present study was to check the consistency among the altimetric retrievals from different GNSS signals. Under this context, the fitting method would require a proper tuning for each case along the leading edge of the waveform, thus causing an additional degree of uncertainty. Therefore, we decided to select the derivative method for the present analysis due to its simplicity and for the sake of a fairest comparison between different GNSS signals, but applied to both data and model power waveforms (differential approach).

For every pair of data and model power waveforms, we computed the residual altimetric delay ($\Delta \rho$) as:

$$\Delta \rho[t,s] = \rho_{\text{spec}}^{\text{data}}[t,s] - (\rho_{\text{spec}}^{\text{model}}[t,s] - \rho_{\text{ecc}}[t,s] + \rho_{\text{trop}}[t]), \tag{4}$$

where $\rho_{\text{spec}}^{\text{data/model}}$ are the specular delays estimated from data and model power waveforms, and the second term has to be corrected by the respective contributions from eccentricity $\rho_{\text{ecc}}$ and tropospheric $\rho_{\text{trop}}$ delays defined in the previous Section 3.2. Then, we proceeded to a general calibration step. The overall calibration was done by combining the results from all available GNSS signals and linearly fitting the evolution of $\Delta \rho$ as a function of sinus of the elevation angle ($\varepsilon$):

$$\Delta \rho = 2\Delta H \sin(\varepsilon) + K_{\text{inst}}, \tag{5}$$

where $\Delta H$ accounts for the average height (surface-to-receiver) difference between data and model, i.e., an overall mismodeling in the vertical direction. The constant term $K_{\text{inst}}$, with no dependency on $\varepsilon$, was assumed as a calibration instrumental delay offset which had to be removed prior to the altimetric inversion. We had to take into account that this approach is valid in this context given that most of the along-track expected variations are caused by the height gradients present in the geoid model (corrected when applying Equation (4)).

It is important to point out that the analysis of $\Delta \rho$ is the proper mean to evaluate the quality of the GNSS-R dataset, given that the altimetric inversion is subjected to additional uncertainties, such as limitations of models for the interpretation of geophysical parameters.

Finally, for every sample, we can compute a corrected version of the specular delay from data ($\rho_{\text{spec}}^{\text{data}}$) as:

$$\widehat{\rho_{\text{spec}}^{\text{data}}}[t,s] = \Delta \rho[t,s] + \rho_{\text{geo}}^{\text{FIN}}[t,s] - K_{\text{inst}}, \tag{6}$$

where $\rho_{\text{geo}}^{\text{FIN}}$ stands for the geometric delay computed over Finnish N2000 height system. The difference between the model terms contained in $\Delta \rho$ (last three terms in Equation (4)) and $\rho_{\text{geo}}^{\text{FIN}}$ was interpreted as

the delay correction predicted by our waveform model (caused by the effects mentioned at the beginning of previous Section 3.3) after subtracting its corresponding input height information. Therefore, $\widehat{\rho_{\text{spec}}^{\text{data}}}$ is just $\rho_{\text{spec}}^{\text{data}}$ corrected by model and calibration terms without any a-priori knowledge of the sea surface level.

### 3.5. Altimetry Inversion

We inverted the altimetric solutions of the sea surface, referred as sea surface heights (SSH), from the retrieved specular delays of every sample as:

$$\text{SSH}_{\text{data}}[t, s] = (\rho_{\text{geo}}^{\text{WGS84}}[t, s] - \widehat{\rho_{\text{spec}}^{\text{data}}}[t, s]) / 2 \sin(\varepsilon[t, s]). \tag{7}$$

Essentially, we obtained the difference between the geometric delay defined in previous Section 3.2 and the corrected version of the specular delay from data from Equation (6), and then we inverted such excess delay into a height value by applying the known sinus of elevation rule. Given that $\rho_{\text{geo}}^{\text{WGS84}}$ was computed over the reference ellipsoid WGS84, the resultant $\text{SSH}_{\text{data}}$ was interpreted as a differential height with respect to such reference (ellipsoidal height), and thus can be compared with the reference sea surface height estimation obtained by means of ancillary measurements ($\text{SSH}_{\text{ref}}$ from previous Equation (1)).

## 4. Results and Discussion

### 4.1. Preliminary Considerations

Before entering into discussion of the altimetric results, Figure 5 provides the time evolution of the normalized power waveforms (data and models) for different GNSS transmitters. In order to achieve a more effective representation, the tracks were connected in the temporal domain by following a sequential order from flight segment 0-BA until 8-DC. Then, the variable in the horizontal axis was understood as the relative time with respect to the first sample in segment 0-BA after removing inter-segment gaps (or time from connected segments). Its correspondence with the actual GPS time is given in Table 1. For practical reasons, the integration time in the present analysis was set to 10 ms (coherent) and 10 s (incoherent). As explained in [33], we found that such coherent integration offered a proper trade-off between SNR improvement and number of independent waveforms for the incoherent averaging. On the other hand, the 10 s interval provided enough reduction of speckle impact for a decent matching between the shape of data and model power waveforms. Moreover, in terms of time integration dependence of the altimetric errors, the results are consistent with those obtained in [23], so there was no added value on this respect.

Back to the waveform time-series, we can see that there was a general good agreement between data and model, as better illustrated in panels showing their difference (in absolute value). By being normalized with respect to their maximum value, we can clearly distinguish different tracks by means of floor noise variations. This effect was mainly caused by changes of the aircraft's orientation with respect to the GNSS transmitters' locations, which modifies the array factor, and it was properly estimated by our model. However, there were two particular cases that show important discrepancies: tracks from Galileo PRN19 and Galileo PRN11 during segment 6-BA. In these situations, the cause is explained by contamination from other GNSS signals, which happens when the proper separability between them cannot be achieved. We illustrate this effect in Figure 6 by comparing the simulated projection of a set of relevant parameters (range, antenna gain and Doppler) over the sea surface for two different cases: clean and contaminated. The corresponding normalized power waveforms, both for model and data, are also shown. We can see how, while in the clean case the main beam of the antenna mostly covered the desired GNSS contribution (Galileo PRN19) marked by its iso-range lines, additional GNSS signals were collected in the contaminated case, whose main beam was expanded by the limitations of the antenna arrays at lower elevations due to its parallel orientation with respect to

the sea surface (best performance achieved at zenith/nadir). The shape of the data waveforms obtained, compared against their corresponding single-GNSS models, clearly shows a different behavior in both cases, where it is not difficult to identify in which situation there are additional contributions.

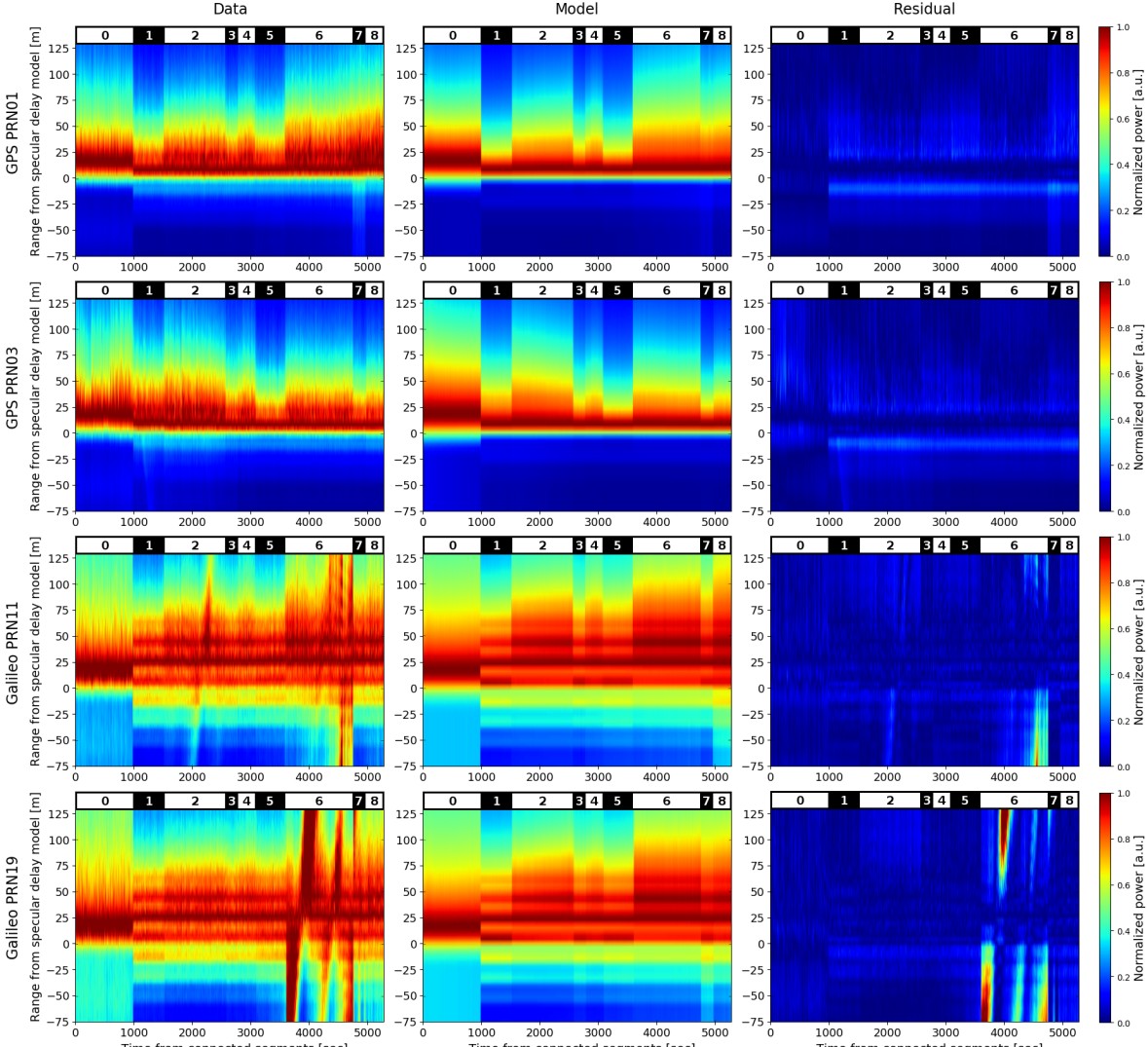

**Figure 5.** **Left**- to **right**-side: normalized power waveforms time-series from data, model and their difference (in absolute value). **Up** to **bottom**: GPS PRN01, GPS PRN03, Galileo PRN11 and Galileo PRN19. The markers on top of each panel indicate the intervals of the different tracks with their corresponding flight segment number.

In spite that we could try to model these contaminated scenarios in order to correct the altimetric distortions produced, we considered that such effort were not worthwhile given that a spaceborne mission will be properly designed to avoid this type of situations, as analyzed in [34]. We have to take into account that the geometry characteristics from a spaceborne instrument offer enhanced separability both in range and Doppler domain (higher delays and receiver velocity); so, if we also include the directivity improvement achieved by employing antenna arrays with an increased number of elements, the resultant scenario is clearly more robust against GNSS cross-contamination. The approach followed here was simply to filter track from Galileo PRN19 during segment 6-BA out of the analysis, which shows a stronger impact around the specular delay location than the equivalent track from Galileo PRN11.

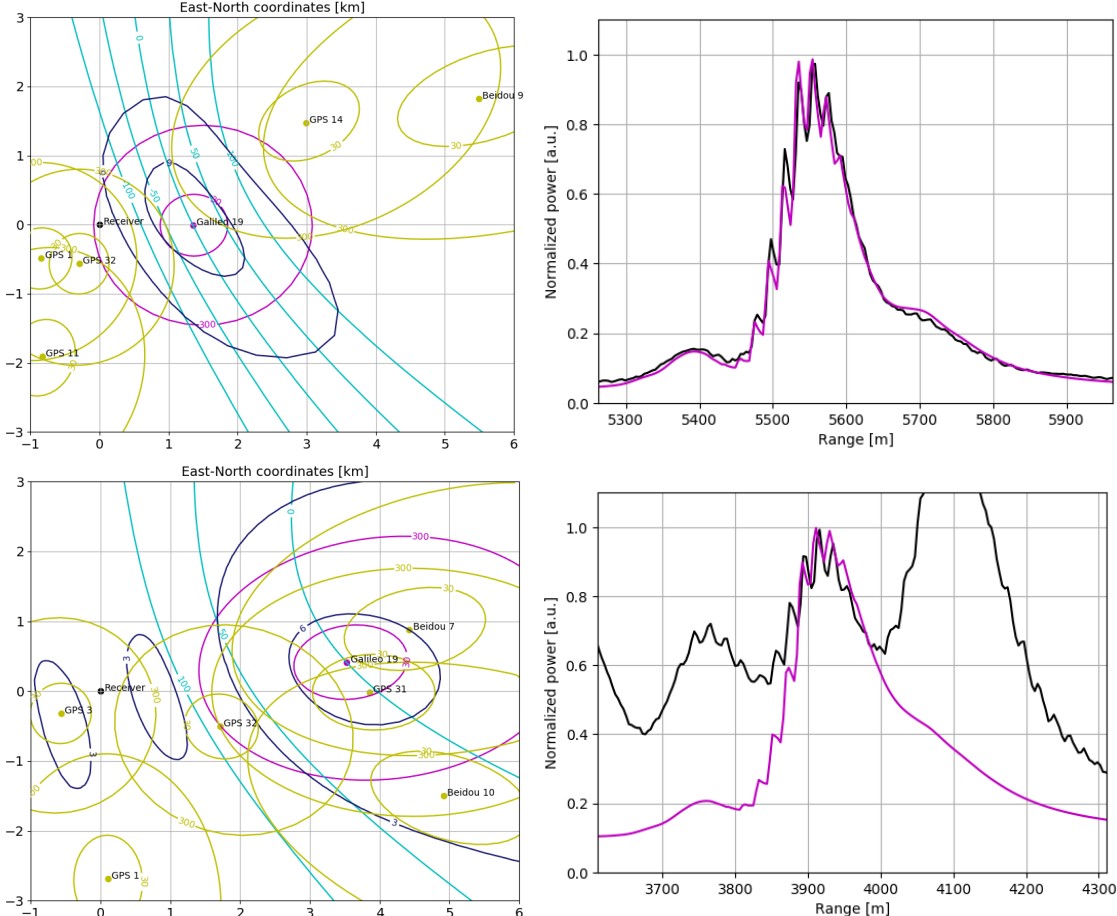

**Figure 6.** (**Up**) Clean example case for Galileo PRN19 at GPS second of the day (SoD) 39,200 (second #1070 in time from connected segments). (**Bottom**) Contaminated example case for Galileo PRN19 at GPS SoD 43,800 (second #4110 in time from connected segments). (**Left-side**) Simulated projection of iso-range (magenta color and meter units), iso-Doppler (cyan color and Hz units) and down-looking antenna array gain (dark blue color and dB units) contour lines over the sea surface for Galileo PRN19 (the origin is set to the vertical projection of the receiver's location). The corresponding iso-range contours of additional GNSS signals in view are also plotted (yellow color and meter units). (**Right-side**) Normalized power waveform for data (black) and corresponding model assuming a single contribution from Galileo PRN19 (magenta).

### 4.2. Specular Delay Analysis

The dataset under analysis was composed of 2000 samples of 10 s of integration, which would be equivalent to a single track of more than five and a half hours of duration. By combining the contributions from eight GNSS signals (four satellites—GPS PRN01, GPS PRN03, Galileo PRN11 and Galileo PRN19—and two frequency bands—L1/L5), an elevation range between ∼28° and ∼83° is covered. For every track, we apply a method for the removal of outliers in the time series of $\Delta\rho$, consisting in filtering out those samples outside a $\pm 2$ standard deviation margin around the mean value, which causes a reduction of 4.8% of the total dataset.

The left-side panel from Figure 7 provides the evolution of $\Delta\rho$ as a function of $\sin(\varepsilon)$ and the result of the linear fit from Equation (5). The values obtained for $\Delta H$ and $K_{\text{inst}}$ are $-0.60$ m and 0.77 m respectively, with 0.05 m and 0.04 m of standard deviation. The first term was interpreted as the differential receiver's height between data and model. By being a negative value, it means that the surface level sensed by the data was, on average, above the surface height of the model (set to Finnish N2000 height system) by 0.60 m. At a first glimpse, we can see that there is consistency among the results from the different GNSS signals and that a straight line properly describes their

combined evolution. This impression is confirmed by the estimated probability density function of each case with respect to the linear fit, that are also shown in the right-side panel from the same figure. Their corresponding statistical moments are provided in Table 2. Although there was not a perfect alignment, which reveals that a finer tuning of the models is still required, the maximum distance between the mean values was just 0.37 m, which was similar to the minimum standard deviation obtained (0.35 m for GPS PRN01 at L1). Such differences in the mean values were behind the rise of skewness (in absolute terms) and kurtosis in the combined case.

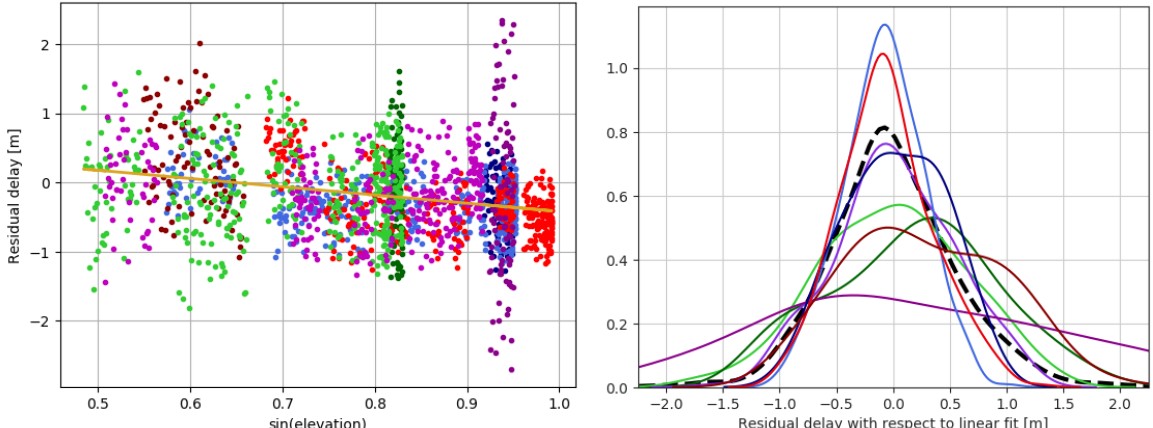

**Figure 7.** (**Left**) Evolution $\Delta\rho$ as a function of $\sin(\varepsilon)$. The different color stands for different PRN: GPS PRN01 in blue, GPS PRN03 in red, Galileo PRN11 in green and Galileo PRN19 in magenta. A darker tone refers to L5 results. (**Right**) Estimated probability density function of $\Delta\rho$ with respect to the linear fit for the different GNSS signals (same color-code) and for their combination (black dashed line).

Finally, based on the results obtained, we replace $K_{\mathrm{inst}}$ in Equation (6) by a signal dependent $K_{\mathrm{inst}}[s]$, whose values are computed as the sum of $K_{\mathrm{inst}}$ (the main contributor) with the projection of the corresponding mean values given in Table 2 in the vertical axis of the evolution of $\Delta\rho$ as a function of $\sin(\varepsilon)$ (in order to keep them independent from the slope information). This approach can be understood as a final refinement of the specular delay bias (without height information) based on the particularities of each type of GNSS signal (for example, the power weight of each code in the composite autocorrelation function). Note that, with the exception of GPS L5, the best agreement is found between PRN's of the same system and frequency band.

**Table 2.** Statistical moments of $\Delta\rho$ with respect to the linear fit for each global navigation satellite system (GNSS) signal and their combination. Integration time applied: 10 ms (coherent) and 10 s (incoherent).

| | Signal | | Mean | $\sigma$ | Skewness | Kurtosis |
|---|---|---|---|---|---|---|
| GNSS | PRN | Band | [m] | [m] | | |
| GPS | 01 | L1 | −0.09 | 0.35 | −0.08 | −0.19 |
| | | L5 | 0.02 | 0.43 | −0.11 | −0.82 |
| | 03 | L1 | −0.06 | 0.41 | 0.26 | −0.02 |
| | | L5 | 0.28 | 0.67 | 0.15 | −0.78 |
| Galileo | 11 | L1 | −0.01 | 0.65 | −0.11 | −0.28 |
| | | L5 | 0.18 | 0.71 | −0.06 | −0.60 |
| | 19 | L1 | 0.01 | 0.53 | −0.06 | −0.25 |
| | | L5 | 0.17 | 1.24 | 0.19 | −0.72 |
| All signals | | | 0.00 | 0.58 | 0.37 | 1.63 |

### 4.3. Altimetry Analysis

The time evolution of our estimated ellipsoidal heights (SSH$_{data}$) and their corresponding values of both Finnish N2000 height system (the a-priori surface from our waveform model) and the in-situ sea surface height reference (SSH$_{ref}$) are displayed in Figure 8. In most of the cases, the results follow the height gradients shown by the reference, as it can be better seen from the linear fits applied to data tracks. In terms of absolute values, the data retrievals are at a similar height level as the ground truth, being both above the values from Finnish N2000 height system, which does not account for effects such as the sea tide level. Note that this is in good agreement with the interpretation of the obtained $\Delta H$ in previous Section 4.2.

In terms of precision, Tables 3–6 provide the standard deviation of SSH$_{data}$ ($\sigma_{SSH}$) with respect to their linear fits for the different tracks and PRN. In order to evaluate how they match with the sea surface height reference, the statistics of the difference between SSH$_{data}$ and SSH$_{ref}$ ($\Delta$SSH) are also provided (the overall combined result has a mean of 0.01 m and a standard deviation of 0.40 m). The list of terms includes two additional variables: signal to noise ratio (SNR) and sensitivity (*S*), both averaged and computed at the specular delay location. We define the last term as the ratio between the waveform's power value and its first range-derivative and, according to [19], it is directly proportional to $\sigma_{SSH}$. In general terms, the values of $\sigma_{SSH}$ are consistent with the evolution of both SNR and *S*. Such relation explains why GPS performs better than Galileo, which transmitted lower power (still not operational during the campaign) and had an autocorrelation function with a less steep rising edge around the specular delay (higher *S*). Moreover, the improvement of L1 with respect to L5 or the sequential degradation with track evolution due to the decrease of the elevation angle (in reverse sense for GPS PRN3) are also a consequence of the same reason. To establish a direct determination of theoretical $\sigma_{SSH}$ from SNR and *S* is not straightforward in this particular scenario, with different GNSS signals and aircraft velocities affecting the coherence time of the measurements (correlation time between consecutive waveforms). For example, tracks with similar values of SNR and *S* showed better results for higher receiver's velocity. This is due to the fact that, under the same integration interval, a higher velocity allows a more efficient speckle noise reduction (larger number of independent samples). A comprehensive analysis on the characterization of the altimetric precision from the waveform statistics is already given in [33], therefore this task is out of scope of the present paper. Regarding the results of $\Delta$SSH, we can see that, while there is an almost perfect matching between $\sigma_{SSH}$ and $\sigma_{\Delta SSH}$, the variations in the mean values indicate that some inter-track offsets still remain. This behavior reveals that, rather than noise-limited, our dataset had residual non-modelled effects (including those belonging to the estimation of SSH$_{ref}$), that, once corrected, could reach the best altimetric performance for this type of system and scenario conditions. One example is the estimation of the Electromagnetic Bias [35], which has not been included in this analysis.

We will now evaluate the consistency between different tracks and GNSS signals, which is the main target of this study and represents an added value for GNSS-R altimetry. The left-side panel from Figure 9 illustrates the projection of the linear fits applied to SSH$_{data}$ over a map with the contour lines of Finnish N2000 height system plus the mean value of in-situ sea level estimation. The height gradient is consistently followed by the fitted retrievals obtained. There are some minor disagreements in the family of CD/DC tracks, corresponding to the cases at the lowest elevations. We can also notice this effect in previous Figure 8. However, taking into account the similar values obtained for $\sigma_{SSH}$ and $\sigma_{\Delta SSH}$, we can assume that these results are caused by the limitation of the linear fit approach in these particular cases, where the length of the track is short compared to the height dispersion. In spite of that, by computing the height differences of the linear fits at the cross-over points, we obtain a mean square difference of just 0.19 m. The right-side panel of Figure 9 provides the estimated probability density function of $\Delta$SSH for each GNSS signal. The results obtained basically confirms the previous conclusions: on one hand there is a good consistency among different GNSS signals, given that the separation between the mean values of these functions, ranging between 0.01 and 0.26 m, is similar to the minimum standard deviation obtained (0.23 m for GPS PRN01 at L1); but on the other hand some

detailed refinement of the models would be required to reach a last minor improvement and getting a perfect alignment (e.g., a better knowledge of the power distribution of the code components from the different GNSS signals).

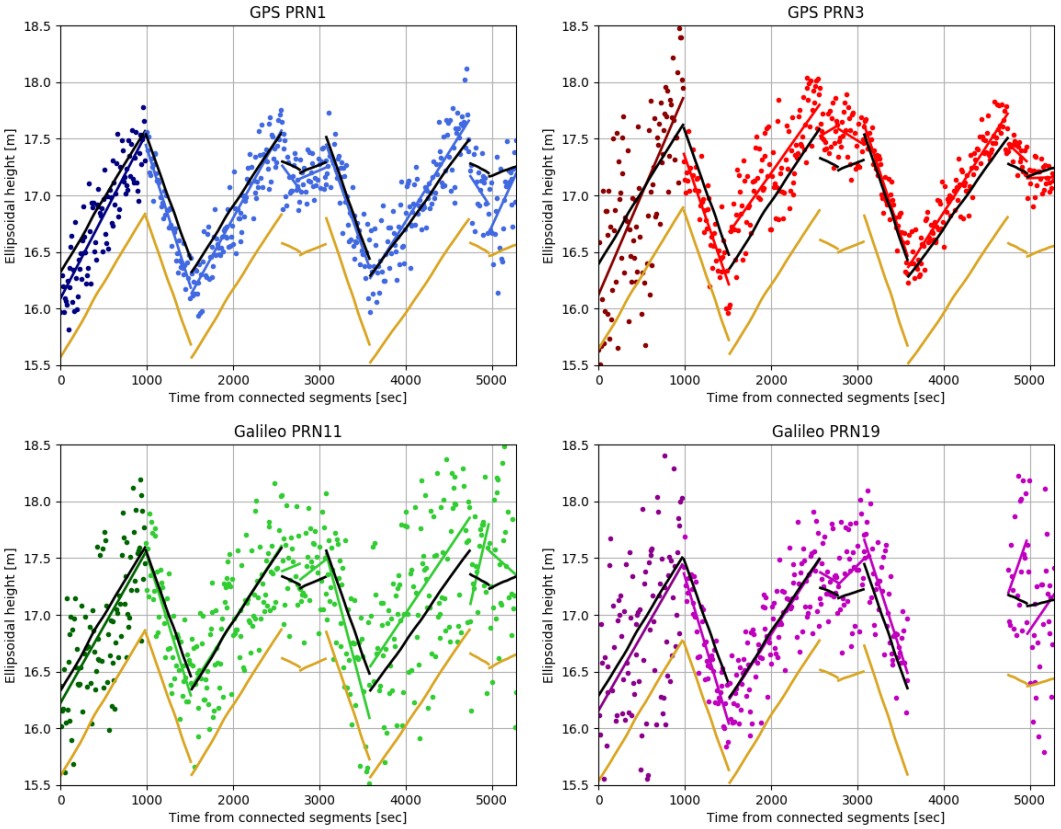

**Figure 8.** Time-series of $SSH_{data}$ (dots) for each PRN and frequency band (a darker tone is employed for L5). A solid line of the same color indicates the linear fit applied for each track. The corresponding values of $SSH_{ref}$ (black) and Finnish N2000 height system (gold) are also plotted.

**Table 3.** Altimetric results for GPS PRN01 with an integration time of 10 m (coherent) and 10 s (incoherent).

| Segment Label | $\sigma_{SSH}$ [m] | $\overline{\Delta SSH}$ [m] | $\sigma_{\Delta SSH}$ [m] | $\overline{SNR}$ [dB] | $\overline{S}$ [m] |
|:---:|:---:|:---:|:---:|:---:|:---:|
| 0-BA | 0.23 | −0.14 | 0.23 | 14.6 | 6.0 |
| 1-AB | 0.12 | −0.17 | 0.13 | 15.5 | 4.0 |
| 2-BA | 0.18 | −0.09 | 0.18 | 15.2 | 4.2 |
| 3-CD | 0.19 | −0.11 | 0.19 | 14.9 | 4.2 |
| 4-DC | 0.14 | −0.05 | 0.14 | 15.6 | 4.2 |
| 5-AB | 0.20 | −0.07 | 0.20 | 13.2 | 4.1 |
| 6-BA | 0.22 | 0.08 | 0.22 | 12.5 | 4.3 |
| 7-CD | 0.22 | −0.20 | 0.23 | 6.3 | 5.8 |
| 8-DC | 0.30 | −0.27 | 0.33 | 10.0 | 5.0 |

**Table 4.** Altimetric results for GPS PRN03 with an integration time of 10 ms (coherent) and 10 s (incoherent).

| Segment Label | $\sigma_{SSH}$ [m] | $\overline{\Delta SSH}$ [m] | $\sigma_{\Delta SSH}$ [m] | $\overline{SNR}$ [dB] | $\overline{S}$ [m] |
|---|---|---|---|---|---|
| 0-BA | 0.53 | −0.03 | 0.56 | 11.3 | 6.4 |
| 1-AB | 0.20 | −0.24 | 0.20 | 10.7 | 4.7 |
| 2-BA | 0.28 | 0.26 | 0.28 | 11.3 | 4.5 |
| 3-CD | 0.15 | 0.28 | 0.16 | 12.2 | 3.9 |
| 4-DC | 0.19 | 0.26 | 0.20 | 14.3 | 3.9 |
| 5-AB | 0.12 | 0.06 | 0.12 | 15.3 | 3.7 |
| 6-BA | 0.15 | 0.15 | 0.15 | 14.6 | 3.9 |
| 7-CD | 0.09 | 0.14 | 0.09 | 14.6 | 3.7 |
| 8-DC | 0.11 | −0.05 | 0.11 | 15.0 | 3.9 |

**Table 5.** Altimetric results for Galileo PRN11 with an integration time of 10 ms (coherent) and 10 s (incoherent).

| Segment Label | $\sigma_{SSH}$ [m] | $\overline{\Delta SSH}$ [m] | $\sigma_{\Delta SSH}$ [m] | $\overline{SNR}$ [dB] | $\overline{S}$ [m] |
|---|---|---|---|---|---|
| 0-BA | 0.43 | −0.09 | 0.43 | 0.4 | 9.0 |
| 1-AB | 0.27 | −0.09 | 0.28 | 5.2 | 7.4 |
| 2-BA | 0.36 | 0.01 | 0.36 | 10.0 | 9.3 |
| 3-CD | 0.31 | 0.11 | 0.31 | 7.9 | 7.8 |
| 4-DC | 0.30 | 0.11 | 0.30 | 10.3 | 7.3 |
| 5-AB | 0.42 | −0.20 | 0.43 | 7.7 | 8.4 |
| 6-BA | 0.60 | 0.25 | 0.60 | 8.7 | 12.2 |
| 7-CD | 0.50 | 0.14 | 0.56 | 8.7 | 9.5 |
| 8-DC | 0.69 | 0.17 | 0.69 | 6.3 | 8.7 |

**Table 6.** Altimetric results for Galileo PRN19 with an integration time of 10 ms (coherent) and 10 s (incoherent).

| Segment Label | $\sigma_{SSH}$ [m] | $\overline{\Delta SSH}$ [m] | $\sigma_{\Delta SSH}$ [m] | $\overline{SNR}$ [dB] | $\overline{S}$ [m] |
|---|---|---|---|---|---|
| 0-BA | 0.66 | −0.10 | 0.66 | 0.5 | 10.9 |
| 1-AB | 0.20 | −0.23 | 0.21 | 10.7 | 7.2 |
| 2-BA | 0.28 | −0.02 | 0.28 | 13.2 | 8.3 |
| 3-CD | 0.23 | 0.12 | 0.23 | 7.7 | 7.2 |
| 4-DC | 0.35 | 0.19 | 0.35 | 8.4 | 8.0 |
| 5-AB | 0.29 | 0.18 | 0.29 | 6.0 | 9.0 |
| 6-BA | - | - | - | - | - |
| 7-CD | 0.44 | 0.29 | 0.47 | 5.0 | 10.4 |
| 8-DC | 0.64 | −0.09 | 0.65 | 2.8 | 11.7 |

The last exercise consists of analyzing the two families of tracks by dividing them into two groups depending on the flight path. The first one takes into account the AB/BA tracks, where there is a significant ellipsoidal height gradient. Then, the values of $SSH_{data}$ are linearly fitted as a function of the longitude dimension from the specular point's location. Similarly, the second group reunites the CD/DC tracks and the same type of computation is done but as a function of the latitude dimension, where there is a nearly flat ellipsoidal height variation. The solutions obtained compared against the corresponding averaged tracks of $SSH_{ref}$ are shown in Figure 10. As expected from previous results, there is good agreement between them.

Finally, it is worth mentioning the implications of these results for a possible spaceborne scenario. All the previous exercises have shown the consistency of a set of altimetry retrievals from a wide variety of GNSS signals and elevations. The relatively low height of the receiver has facilitated this task because the resultant specular points are concentrated over the same monitorization area. Under a spaceborne scenario, the distance between these observations would significantly increase, thus allowing the altimetric retrieval from separated areas at the same time to complement monostatic

Radar measurements. However, intrinsic biases or other non-controlled effects could have a different impact on each observation depending on the type of signal or its geometry. In this context, the outcome from the present work serves as a preliminary proof of the robustness of the iGNSS-R concept towards multistatic altimetry.

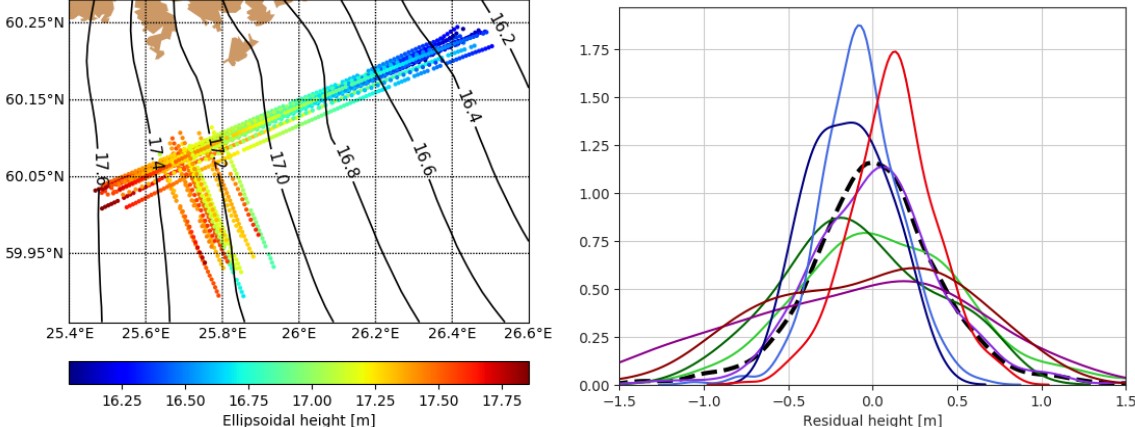

**Figure 9.** (**Left**) Results from the linear fits applied to $SSH_{data}$ located at their corresponding specular points. The contour lines of Finnish N2000 height system plus the mean value of in-situ sea level estimation are included (black lines and meter units). (**Right**) Estimated probability density function of $\Delta SSH$ for each GNSS signal (same color code as in previous Figure 8) and for their combination (black dashed line).

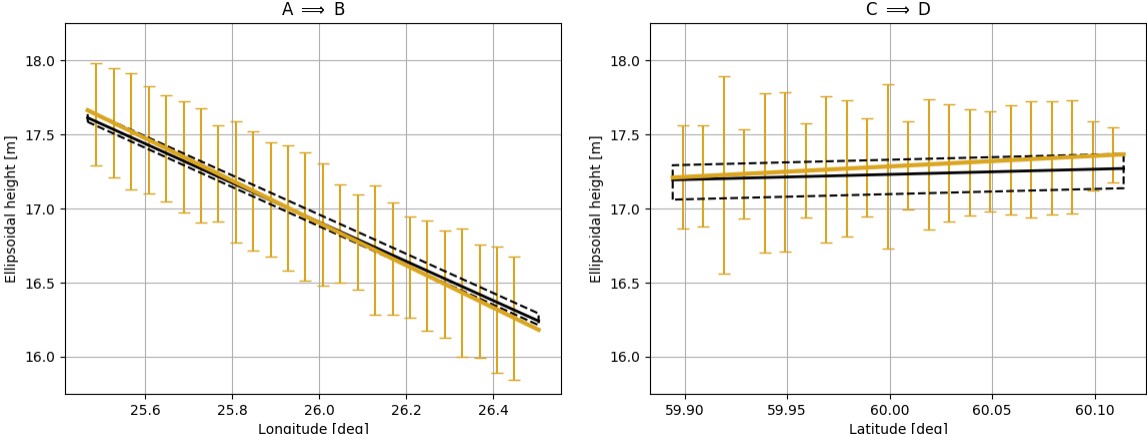

**Figure 10.** (**Left**) Linear fit of $SSH_{data}$ results from the first group (AB/BA) as a function of longitude component from their location (gold color). The error bars indicate $\pm\sigma$ intervals along the fitted track. The corresponding results of $SSH_{ref}$ are also plotted (black), where the dashed lines indicate the range of values of $SSH_{ref}$ along the track (instead of $\pm\sigma$ intervals). (**Right**) The equivalent result of the second group (CD/DC) as a function of latitude.

## 5. Conclusions

The main target of the present study has been to evaluate the capability of iGNSS-R to perform consistent and simultaneous altimetry measurements over a broad area (synoptic scenario). The relevance of such goal lays in the fact that the increasing GNSS constellation represents a unique advantage of this technique. We could imagine the enhanced spatial coverage that might be achieved with such a wide variety of available transmitters from a spaceborne platform. Therefore, the capability of employing different signals towards GNSS-R synoptic altimetry is a key aspect to complement conventional missions based on Radar and SAR, which on the other hand are benefited with dedicated signals with higher power and bandwidth.

After proving, for the first time, the interferometric concept and assessing its performance in terms of altimetry precision [22,23], we have developed a new instrument with beamformer capability to assess whether it is possible to achieve accurate synoptic altimetry with iGNSS-R. For such purpose, a new aircraft campaign was then carried out to acquire GNSS-R signals with the aim of checking the consistency and reliability of their altimetry retrieval in a known scenario. This study presents the results obtained in this campaign.

The results obtained show consistency among the whole dataset for a wide elevation range, which combines two GNSS systems and two frequency bands, with discrepancies between 0.01 and 0.26 m. Moreover, the absolute sea surface height estimations are in good agreement with in-situ measurements, with an overall standard deviation of height residual of 0.40 m, following a clear height gradient predicted by the reference surface model. The error levels are in accordance with the particular characteristics of each GNSS signal, ranging between 0.09 and 0.66 m depending on the particular case for an integration of 10 s.

The dataset analyzed here is the first one that permits the evaluation of multiple aspects of the accuracy of iGNSS-R altimetry: by comparing with reference surface information and by comparing individual tracks from different GNSS transmitters and geometries. The consistency shown by the results represents a key aspect towards the assessment of the iGNSS-R concept for a possible spaceborne mission, where the spatial separation of the specular points would allow the monitorization of mesoscale features over the ocean.

Despite that the followed procedure incorporates delay corrections from a comprehensive waveform model, the results obtained reveal some residual differences between tracks, which indicates that more effort is required to properly model all systematic effects. However, in view of an hypothetic spaceborne mission, a better option should be to solve these issues in a first stage by means of calibration and validation measurements over specific sites characterized for such purpose.

Finally, there is still room of improvement for those cases that exhibit larger errors. Firstly, the worse behavior at lower elevations could be mitigated by improving the beamforming strategy (e.g., increasing the number of antenna elements or changing their orientation towards slant geometries). Secondly, Galileo signals were not yet operational during the experimental campaign and they had lower power levels than nominal. As a last point, as we shown in [33], there is a potential improvement by determining the specular delay by means of a fitting procedure. However, as it has been already discussed, such method is not adequate for the sake of a fair comparison between different GNSS signals, which is the main purpose of this paper.

**Author Contributions:** The signal processor and data-analysis studies were done by F.F., W.L., A.R. and E.C. The SPIR instrument was designed and built by S.R., J.C.A.-F. and O.N.-C. The experimental campaigns were planned and carried out by S.R., F.F., J.P., E.R., J.S. and M.M.-N.

**Funding:** This work has been carried out with the financial support of the following research projects and contracts: Spanish research grant ESP2015-70014-C2-2-R (MINECO/FEDER) and ESA Contract: RFQ/3-12747/09/NL/JD-CCN4.

**Acknowledgments:** Simulation of waveform model, specular point computation, beamformer's phases determination and all the data analysis described in this work have been made with wavpy [36], our open-source library available for the scientific community. Most of the figures (with the only exception of Figure 2) have been made with Matplotlib [37], a Python graphics package.

**Conflicts of Interest:** The authors declare no conflict of interest.

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
