# Peer review of "Is Accurate Synoptic Altimetry Achievable by Means of Interferometric GNSS-R?"

_remotesensing, doi:10.3390/rs11050505_

Round 1

Reviewer 1 Report

Please see attached

Author Response

@page { margin: 0.79in } p { margin-bottom: 0.1in; line-height: 120% }

The paper investigated the performance consistency of interferometry GNSS-R over a broad area with data related to different satellites and different frequency bands. The paper is well written and organized in general with detailed analysis and extensive experimental results. A number of minor comments are as follows.

Thanks for your comments.

1) Line 30, it is better to add “(i.e. SIR-C and SMAP)” following “Although the aforementioned spaceborne systems”

R1) Here we are referring to TDS-1 and CYGNSS.

2) Line 68, “at a different geometries” should be “at a different geometry” or “at different geometries”

R2) Right, the correction is applied (line 68).

3) Lines 85-89, it is best to tell reader (in the first paragraph of section 2) what “SPIR” means and what the SPIR experimental campaign is

R3) The first paragraph has been moved to the end of the section as a summary of the relevant points (lines 154-159). Then, the SPIR concept is now properly presented.

4) Section 3.1, it is useful to add a flow chart from raw signals to power waveforms in section 3.1 is more favorable

R4) The flow chart has been added (figure 4).

5) Lines 188-189, equation (2), explain why the second term is subtracted and the third term is added

R5) It depends on their definition with respect to the first term and we have remarked this aspect in the new version (lines 196 and 206). The second term is subtracted because the reflected signal reaches the down-looking array location before than the reference point (assigned to the up-looking array). The third term is added because accounts for the excess delay caused by the troposphere and it is not taken into account by the first term, which is purely geometrical.

6) Section 3.3, it is best to add an example of waveform model and the explanation of the example for simulated and observed data

R6) Indeed, detailed model-data comparisons would better illustrate the impact of the effects mentioned here. However, in addition that this is out of the scope of the paper, there are two main problems: (1) the scale of the effects (meter level) is rather small compared against the size of the waveforms (>100 meter just taking the most significant interval), and (2), the number of degrees of freedom in this dataset (two bands and two systems) would require too many cases to achieve a fair analysis. On the other hand, examples of waveform models and their comparison against observed data were given in former figures 4 and 5 (5 and 6 in the new version).

7) Line 223-224, “Our approach is based on [29] and includes the coherent signal contribution from [30] with expressions in [31]”, it is best to explicitly mention based on what and from what

R7) Right, we have clarified this point (lines 225-226).

8) Line 242, “of” is missing in “along the leading edge the waveform”

R8) Thanks. Corrected (line 244).

9) Line 256, “to” is missing in “prior the altimetric inversion.”

R9) Thanks. Corrected (line 257).

10) Line 263-264, it is useful to give more explanations on how equation (6) is obtained

R10) The purpose of equation (6) is to extract the instrumental delay Kinst obtained after applying the linear fit from equation (5) and to remove the height information contained in ρmodelspecWe have added a sentence to clarify this aspect (lines 270-271).

11) Line 312-313, “we consider that such effort is not worthwhile given that a spaceborne mission will be properly designed to avoid this type of situations”, it may not be easy to avoid this type of situations

R11) Right. A reference has been added on this respect (line 317).

12) Line 327, “which causes a reduction of 4.8% of the total dataset”, it is better to mention that in practice, exclusion of outliers is usually applied in the post-processing of data

R12) Yes, it is an standard procedure. However, +/-2 σ is sometimes considered a too tight margin, so we need to provide this information.

13) Line 328, “the top panel” should be “the left panel”

R13) Thanks. Corrected (line 332).

14) Line 332, since it is mentioned on line 253 that “DH accounts for the average height difference between data and model”, in “the surface level sensed by the data is, on average, above the surface height of the model”, “above” should be replaced with “below”

R14)ΔH accounts for surface-to-receiver heights, which go in opposite sense thasea surface heights. We have clarified this point after equation (5) (line 255).

15) Line 336, “in the bottom panel” should be “in the right panel”

R15) Thanks. Corrected (line 340).

16) Page 14, Tables 3-6, the notations “ ΔSSH ” and “σΔSSH” are confusing; audience would think the latter is the standard deviation of the former; if the former is simply a constant, how can you obtain its standard deviation?

R16) Tables 3-6 provide ΔSSH instead of ΔSSH.We have inserted some vertical space in the tables to better display the horizontal bars.

17) Lines 386-387, “top panel” should be “left panel

R17) Thanks. Corrected (line 390).

Reviewer 2 Report

see attachment.

Author Response

@page { margin: 0.79in } p { margin-bottom: 0.1in; line-height: 120% }

The manuscript evaluates the capability of iGNSS-R in sea surface altimetry of synoptic scenario, focusing on the problem of self-consistency and accuracy of synoptic solutions. The results show that SSHs from different GNSS signals are self-consistent well (discrepancies between 0.01 and 0.26m), and in good agreement with ancillary information (at 0.4m level).

The manuscript focuses on the synoptic scenario of iGNSS-R. A dedicated airborne campaign is designed to collect required dataset. The data is well analyzed, and the results confirm the capability of iGNSS-R in sea surface altimetry. The manuscript is well in writing. The manuscript is recommended for publication after minor revisions.

Thanks for your comments.

Specific comments

(1) This manuscript focused on two issues of iGNSS-R altimetry, self-consistency and accuracy, but the title only mentions accuracy. It would be better if the title covers both issues.

R1) Indeed, self-consistency is a key element of this study. However, we consider that the term synoptic covers this aspect, because for achieving “accurate synoptic altimetry” it means that the different observations must be consistent (single accuracy is not enough).

(2) Line 68: minor grammar error. It should be “several GNSS signals at different geometries ...”, not “several GNSS signals at a different geometries...”.

R2) Thanks. It has been corrected (line 68).

(3) Figure 6 and 8 are left-right displayed, but the context uses the top panel from figure 6 (line 328), the bottom panel (line 336), the top panel from (line 386), and the bottom panel of figure 8 (395). Keep the display of figures in line with the description in context.

R3) Thanks. The description has been corrected (lines 332, 340, 390 and 399).

(4) lines 360-362 mentioned that “...(the overall combined result has a mean of 0.01 m ...”. Tables 3-6 provide the mean values of ΔSSH for different tracks and PRN (the third column of each table). The mean value of combined result is calculated using the values of the third column of tables 3-6, as follows:

Mean={sum([-0.14 -0.17 -0.09 -0.11 -0.05 -0.07 0.08 -0.2 -0.27])+sum([-0.03 -0.24 0.26 0.28 0.26 0.06 0.15 0.14 -0.05])+sum([-0.09 -0.09 0.01 0.11 0.11 -0.2 0.25 0.14 0.17])+sum([-0.1 -0.23 -0.02 0.12 0.19 0.18 0.29 -0.09])}/35=0.016, where 35 is the number of tracks.

This manuscript keeps two decimal fraction, so it is better to turn 0.016 to be 0.02, please the authors check if the mean value of combined result is 0.01.

R4) We compute the overall results by combining all the samples, not track by track. The result obtained this way is 0.0105 m. Note that the number of samples for each track is not constant.

(5) lines 396-399 mentioned that “… given that the separation between their mean values, ranging between 0.01 and 0.26 m, is similar to the minimum standard deviation-0.23 m-;...”. There is a little confused by “their mean values”. According to the near context, the “their mean values” indicates  ΔSSH(the third column of tables 3-6), however, if it is ΔSSH, it should range from 0.01(row 4, column 3 of table 5) to 0.29m (row 9, column 3 of table 6), rather than 0.01 to 0.26m. “the minimum standard deviation -0.23m-” has the similar issue because the minimum standard deviation of ΔSSH is 0.09 according the fourth column of tables 3-6.

Please the authors check what their mean values indicates here, clarify it if it is notΔSSH.

R5) These values refer to the PDF’s from the right-side panel of figure 8, not to tables 3-6. We have clarified this aspect in the text (lines 402-403).